# Effect of Oral Administration of Polyethylene Glycol 400 on Gut Microbiota Composition and Diet-Induced Obesity in Mice

**DOI:** 10.3390/microorganisms11081882

**Published:** 2023-07-26

**Authors:** Riko Ishibashi, Rio Matsuhisa, Mio Nomoto, Seita Chudan, Miyu Nishikawa, Yoshiaki Tabuchi, Shinichi Ikushiro, Yoshinori Nagai, Yukihiro Furusawa

**Affiliations:** 1Department of Pharmaceutical Engineering, Faculty of Engineering, Toyama Prefectural University, 5180 Kurokawa, Toyama 939-0398, Japan; 2Department of Biotechnology, Faculty of Engineering, Toyama Prefectural University, Kurokawa, Toyama 939-0398, Japan; 3Division of Molecular Genetics Research, Life Science Research Center, University of Toyama, Sugitani, Toyama 930-0194, Japan

**Keywords:** polyethylene glycol (PEG), carboxymethyl cellulose (CMC), gut microbiota, obesity, *Parabacteroides goldsteinii*, *Akkermansia muciniphila*

## Abstract

Polyethylene glycol (PEG) is a commonly used dispersant for oral administration of hydrophobic agents. PEG is partly absorbed in the small intestine, and the unabsorbed fraction reaches the large intestine; thus, oral administration of PEG may impact the gut microbial community. However, to the best of our knowledge, no study evaluated the effects of PEG on gut commensal bacteria. Herein, we aimed to determine whether oral administration of PEG modifies the gut microbiota. Administration of PEG400 and PEG4000 altered gut microbial diversity in a concentration-dependent manner. Taxonomic analysis revealed that *Akkermansia muciniphila* and particularly *Parabacteroides goldsteinii* were overrepresented in mice administered with 40% PEG. PEG400 administration ameliorated the high-fat diet (HFD)-induced obesity and adipose tissue inflammation. Fecal microbiome transplantation from PEG400-administered donors counteracted the HFD-induced body and epididymal adipose tissue weight gain, indicating that PEG400-associated bacteria are responsible for the anti-obesity effect. Conversely, carboxymethyl cellulose, also used as a dispersant, did not affect the abundance of these two bacterial species or HFD-induced obesity. In conclusion, we demonstrated that oral administration of a high concentration of PEG400 (40%) alters the gut microbiota composition and ameliorates HFD-induced obesity.

## 1. Introduction

Polyethylene glycol (PEG) 400, a polymer comprising repeating ethylene glycol units, is commonly used as a cosolvent or dispersant for administering hydrophobic agents in animal experiments. Although PEG was once considered inert, recent studies demonstrated that PEG can increase intestinal permeability and inhibit drug efflux transporters and drug-metabolizing enzymes [1,2,3], indicating that oral administration of PEG has the potential to exert systemic effects.

In recent decades, the beneficial and pathogenic effects of commensal gut bacteria, particularly those inhabiting the large intestine, attracted growing attention. The diversity and composition of gut bacteria are reportedly altered in response to changes in the environment, food, and drug administration, affecting the systemic functions of host organisms, such as metabolism [4]. Following oral administration, PEG is partly absorbed by the small intestine, and the remaining unabsorbed portion reaches the large intestine [5]. Considering the abundance of gut bacteria in the large intestine and that PEG is reportedly utilized as a carbon source by several bacterial species [6], we hypothesized that orally administered PEG400 could alter the colonic microbial community, thereby altering the phenotype of experimental animals, even in the absence of dispersed drugs. However, to the best of our knowledge, the effects of PEG on gut microbiota remain unexplored.

Among the diverse effects of host–microbiota interactions, the association between metabolic diseases and gut commensal bacteria was intensively examined for several decades. Initially, Firmicutes and Bacteroidetes, dominant phyla in mice and humans, respectively, were identified as obese- and anti-obesity-associated bacteria at the phylum level, respectively [7].

Some Bacteroides spp. were shown to promote gut barrier function and energy metabolism by producing short-chain fatty acids (SCFAs) [8,9]. Improved gut barrier function can alleviate insulin resistance by inhibiting systemic and chronic inflammation induced by the systemic translocation of lipopolysaccharide (LPS) from the gut lumen [10]. *Parabacteroides goldsteinii*, a species belonging to the Bacteroidetes phylum, was shown to ameliorate obesity and diabetes, probably via unknown metabolites, given that pasteurized *P. goldsteinii* loses its anti-metabolic disease effect [11]. In addition, *Bacteroides acidifaciens* and *Parabacteroides distanosis*, classified as Bacteroidetes spp., were implicated in improving metabolic disease through an unknown mechanism [12,13]. Although not a Bacteroidetes phylum, *Akkermansia muciniphila*, a representative of the phylum Verrucomicrobia, was shown to afford beneficial effects against metabolic disease through direct effects mediated via its membrane protein, i.e., Amuc_1100, and indirectly via SCFA production [14,15,16,17,18]. Therefore, PEG administration could impact the metabolic phenotype of experimental animals if it alters the abundance of the gut microbiota associated with metabolic disease.

In the present study, we aimed to evaluate the effect of PEG400 on the diversity and composition of gut microbiota. Furthermore, we attempted to elucidate the relationship between PEG400-associated bacteria and diet-induced obesity.

## 2. Materials and Methods

### 2.1. Animals and Treatments

C57BL6/J mice (male, 4 weeks old) were purchased from Japan SLC (Hamamatsu, Japan). The mice were maintained under specific pathogen-free conditions and a 12 h shift of the light–dark cycle in the animal facility of Toyama Prefecture University, with free access to water and diet. After a one-week acclimation period, mice were fed a high-fat diet (HFD; 60% fat; Research Diet, New Brunswick, NJ, USA) or normal diet (ND; 10% fat; Research Diet) for the indicated weeks. Then, mice were orally administered 100 µL PEG400 (Nacalai Tesque, Kyoto, Japan), PEG4000 (Nacalai Tesque), or saline (Otsuka, Japan) five times per week. PEG was diluted with saline to prepare 4, 10, and 40% PEG solutions. Sodium carboxymethyl cellulose (CMC; Nacalai Tesque) was dissolved in saline and adjusted to 1% w/w for oral administration. The animal care policies in the experiments were approved by the Animal Experiment Ethics Committee of Toyama Prefectural University (The Ethics Committee Approval No. R4-6).

### 2.2. Fecal DNA Isolation

Feces were collected from mice administered PEG for two or four weeks. Feces were immediately frozen in liquid nitrogen and stored in a deep freezer (−80 °C), as storage at room temperature for more than 15 min or in a domestic freezer for more than three days can impact the composition of bacterial taxa in meta 16S analysis [19]. DNA was extracted from a pellet of feces (≤50 mg) using the ZymoBIOMICS DNA Miniprep kit (Zymo Research, Irvine, CA, USA) following the manufacturer’s protocol. Bead beating was performed for 20 min using a Disruptor Genie instrument (Scientific Industries, Bohemia, NY, USA) at maximum speed.

### 2.3. 16S Ribosomal RNA (rRNA) Gene Amplicon Sequencing

For sequencing, a 16S rRNA gene sequence library was prepared according to the Illumina (San Diego, CA, USA) protocol described previously [20,21,22,23]. Briefly, DNA samples were amplified using the KAPA HiFi HS Ready Mix (Roche, Basel, Switzerland) and primers targeting V3 and V4 variable regions of the gene coding 16S rRNA. The resulting PCR products were purified and further amplified using the Nextera XT Index kit (Illumina). The 4 nM pooled libraries in 5 mM Tris-HCl buffer were sequenced using the MiSeq Reagent Kit v3 600-cycle (Illumina).

### 2.4. Bacterial Community Analysis

The raw sequence reads of the V3-V4 regions of the 16S rRNA gene were analyzed using Quantitative Insight into Microbial Ecology 2 (QIIME2) Ver.2021.2 [24]. The primer region (forward, 17 bases; reverse, 21 bases) was trimmed from raw sequences using the Cutadapt plugin. Following paired-end reads joining (forward, 280 bp; reverse, 220 bp), amplicon sequence variants were constructed using the DADA2 algorithm. For diversity analyses, a diversity core-metrics-phylogenetic analysis was used for 5000–20,000 reads set as the sampling depth. For α-diversity, the total number of species in the gut bacterial community (richness) was expressed as “observed features”. For β-diversity, principal coordinates analysis was based on a weighted UniFrac distance matrix. For taxonomic classification analysis, feature classifiers, classify-sklearn, and EZBiocloud databases [25] were used to assign the taxonomy. To calculate relative abundance of taxa, raw read counts were normalized based on total read counts.

### 2.5. RNA isolation and cDNA Preparation

Total RNA was extracted from epididymal white adipose tissue (eWAT) and colonic tissue using a NucleoSpin RNA Mini kit (TaKaRa, Shiga, Japan) following the manufacturer’s procedures. Total RNA (≤1 µg) was reverse transcribed using ReverTra Ace qPCR RT Master Mix (TOYOBO, Osaka, Japan).

### 2.6. Reverse Transcription-Quantitative PCR (RT-qPCR)

RT-qPCR (20 mL/reaction) was performed using cDNA (4 mL/reaction) or fecal DNA (10 ng/reaction), gene- or bacteria-specific primers (0.2 mM/reaction), THUNDERBIRD SYBR qPCR Mix (10 mL/reaction, TOYOBO), and a CFX Connect Real-Time System (Bio-Rad Laboratories, Hercules, CA, USA), according to the instructions provided by manufacturer. The Ct values of the gene or bacterium were normalized with that of Rpl13a or Eubacterium to calculate the relative expression levels of genes or relative abundance of bacteria, respectively. The primer sequences used for qPCR are listed in the Appendix A. The specificity of the primers was already confirmed in our previous study [20].

### 2.7. Insulin Tolerance Test

To measure insulin tolerance, mice were subjected to a 3 h fasting period, followed by the intraperitoneal administration of 0.1 unit/kg insulin (Humulin R, 100 U/mL; Eli Lilly, Kobe, Japan) according to our previous study [20]. At indicated time points post-administration/injection, blood was collected from the abdominal aorta under isoflurane anesthesia. One-mL syringes with 26 G needles were precoated with 0.5 M ethylenediaminetetraacetic acid, followed by blood sampling. Blood glucose levels were measured using a StatStrip Xpress glucose meter (Nova Biomedical, Boston, MA, USA).

### 2.8. Fecal Microbiome Transplantation (FMT)

FMT was performed as previously described [20]. For the bacterial transfer experiments, donor mice received PEG or saline orally for two weeks, and 4–5 fecal pellets (~100 mg) were collected fresh for daily FMT. The collected feces were suspended in 1 mL of phosphate-buffered saline, followed by centrifugation at 2000× *g* for 10 s. To minimize the changes in microbial composition of the supernatant, the 100 μL aliquots of the supernatant containing fecal bacteria were orally administered to recipient mice immediately after centrifugation. Recipient mice were fed a HFD for four weeks, treated with a mixture of four antibiotics for four days and subsequently rested for three days prior to transfer. For antibiotic treatment, an antibiotics cocktail containing ampicillin, neomycin, metronidazole, and vancomycin (Nacalai Tesque) was diluted in sterilized water to final concentrations of 5, 5, 5, and 2.5 mg/mL, respectively. The antibiotics cocktail (200 mL) was orally administered for four days. Oral antibiotic administration via *ad libitum* administration via drinking water was avoided to minimize individual differences in body weight change and antibiotic doses [26]. After three days of rest following antibiotic administration, FMT from donor mice was performed five times weekly for six weeks.

### 2.9. Statistical Analysis

Differences between two groups were analyzed using the Student’s *t*-test or Welch’s test. Differences between more than two groups were analyzed using Tukey’s post-hoc test or the Kruskal–Wallis test. Additionally, *p*-values < 0.05 indicate a statistically significant difference between sample groups.

## 3. Results

### 3.1. Effects of PEG on Gut Microbiota of ND-Fed Mice

To determine the effect of PEG on gut microbiota, we performed 16S rRNA sequencing of fecal samples collected from mice that were orally administered PEG400 or PEG4000 for 2 weeks. To evaluate the dose–response effect, the PEG concentration was increased from 4 to 40%, given that ~40% PEG400 is typically used as a hydrophobic drug dispersant. Firstly, we analyzed α- and β-diversities to compare the number of observed species and the microbial composition between saline- and PEG-administered mice. Diversity analysis revealed that both PEG400 and PEG4000 decreased the number of observed species in a concentration-dependent manner (Figure 1A). In particular, high concentrations (40%) of PEG400 and PEG4000 dramatically reduced the number of bacterial species in the gut. Corroborating the change in α-diversity analysis, the gut bacterial composition of mice administered 40% PEG400 or 4000 markedly differed from that of other groups (Figure 1B). To identify the bacterial taxa altered by PEG administration, we performed a taxonomic classification of 16S rRNA-targeted amplicons. Considering the taxonomic analysis at the phylum level, Firmicutes and Verrucomicrobia were underrepresented and overrepresented, respectively, in mice administered high concentrations of PEG400 and PEG4000 (Figure 1C). In contrast, bacterial composition at the phylum level was comparable between mice administered CMC and in those administered saline (Figure 1C). Considering the taxonomic analysis at the species level, *Akkermansia muciniphila* (phylum: Verrucomicrobia) and *Parabacteroides goldsteinii* (phylum: Bacteroidetes) were overrepresented in mice administered 40% PEG400 and PEG4000 (Figure 1D). Conversely, low concentrations (4 or 10%) of PEG400 and PEG4000 as well as CMC did not markedly increase these two species.

### 3.2. Effects of PEG on Gut Microbiota of HFD-Fed Mice

*A. muciniphila* colonizes the gut mucosal layer and is known to exert beneficial effects against diabetes and obesity [18]. *P. goldsteinii* was recently identified as a novel probiotic candidate for combating metabolic diseases [11]. Therefore, we hypothesized that PEG administration could affect the metabolic phenotype by altering the composition of the gut microbiota. Accordingly, we examined the effect of PEG400 on the gut microbiota and diet-induced obesity in mice fed a HFD, given that PEG400, but not PEG4000, is used as a dispersant for potential drugs against metabolic diseases, such as hyperlipidemia, in animal experiments [27]. Consistent with the gut microbiota composition of PEG-administered ND-fed mice, administering a high concentration of PEG400 reduced the number of species observed in HFD-fed mice (Figure 2A). Based on the β-diversity analysis, high concentrations of PEG400 altered gut microbiota composition, even in mice fed a HFD (Figure 2B). Compared with mice administered PEG for two weeks, mice administered PEG for four weeks exhibited marked alterations in α- and β-diversity (Figure 2A,B). Taxonomic analysis revealed an overrepresentation of *A. muciniphila* and *P. goldsteinii* in mice administered PEG400, even in HFD-fed mice (Figure 2C). However, it should be noted that while marked PEG-induced overrepresentation of *A. muciniphila* was observed in Cage #1, only a slight effect was observed in Cage #2, indicating the so-called cage effect on the bacterial community [28,29]. In contrast, *P. goldstenii* overrepresentation was observed in PEG400-administered mice bred in both Cages #1 and #2; therefore, the PEG-mediated induction of *P. goldstenii* was statistically significant two weeks after administration. These results indicate that *P. goldsteinii*, rather than *A. muciniphila*, is a PEG-responsive bacterium, given that statistically significant differences at two and four weeks were notable even under the influence of the cage effect. Regarding other bacterial species, some Bacteroides spp., including *AB599946*, *PAC002443*, and *Parabacteroides PAC002483*, were also overrepresented in HFD-fed mice administered PEG400 (Figure 2C), although PEG400 decreased AB599946 expression in ND-fed mice (Figure 1D).

### 3.3. Effects of PEG on Appetite and HFD-Induced Body Weight and eWAT Weight Change

Next, we determined whether PEG could ameliorate diet-induced obesity in HFD-fed mice. Mice were fed a HFD and orally administered PEG or saline for eight weeks. We observed that neither 4% nor 40% PEG400 affected appetite, although a high concentration of PEG decreased HFD-induced body weight gain (Figure 2D,E). Consistent with the body weight change, the eWAT weight also differed between mice groups administered 40% PEG400 and saline (Figure 2F).

### 3.4. Effects of PEG on HFD-Induced eWAT Inflammation and Intestinal Barrier Function

PEG400 attenuated the expression of pan-macrophage and WAT M1 macrophage markers F4/80 and CD11c, respectively (Figure 3A). Furthermore, PEG400 decreased the expression of tumor necrosis factor (TNF) a, a pro-inflammatory cytokine secreted from cells such as M1 macrophages (Figure 3A). Conversely, PEG400 elevated the expression of adiponectin (Figure 3A), an adipokine that promotes insulin sensitivity [30]. However, the effect of PEG400 on the recovery of insulin sensitivity was limited, given that significant differences in blood glucose levels between saline- and PEG400-administered mice were observed only 120 min after insulin injection (Figure 3B). Thus, the area under the blood glucose response curve (AUC) did not significantly differ between the two groups (Figure 3C). Moreover, PEG400 administration did not enhance expression levels of *Tjp1* and *Muc2* (Figure 3D), functional markers of the intestinal barrier [31], indicating that PEG400 did not recover gut permeability, regardless of the overrepresentation of beneficial bacteria. Consequently, oral administration of 40% PEG400 altered the gut microbiota composition and ameliorated diet-induced obesity and eWAT inflammation, but not gut barrier dysfunction or insulin resistance.

### 3.5. Effect of Bacterial Transplantation on HFD-Induced Obesity and eWAT Inflammation

To examine the effect of PEG-modified gut bacterial composition on HFD-induced obesity and adipose tissue inflammation, fecal microbiota from saline- or PEG400-treated donor mice were transplanted into bacteria-depleted, HFD-fed recipient mice (Figure 4A). FMT from PEG400-treated donors increased *P. goldsteinii* and *A. muciniphila* abundance in recipient feces, thereby indicating successful transplantation (Figure 4B), and reduced body and eWAT weight gain without impacting appetite (Figure 4C–E). FMT from PEG400-treated donors also decreased inflammatory gene expression and increased adiponectin levels in the adipose tissue (Figure 4F), which was also observed in mice orally administered 40% PEG400 (Figure 2A). These findings indicate that PEG400 could improve HFD-induced obesity and adipose tissue inflammation by modifying the gut bacterial composition.

### 3.6. Effects of CMC on Gut Microbiota, Obesity, and eWAT Inflammation in HFD-Fed Mice

PEG400 can potentially modify the gut microbiota and ameliorate obesity, which may lead to unexpected results in experiments investigating metabolic diseases. Herein, we evaluated the effects of an alternative dispersant, i.e., CMC, on HFD-induced obesity and adipose tissue inflammation. In contrast to 40% PEG400, CMC did not impact the relative abundance of *A. muciniphila* and *P. goldsteinii* in ND-fed mice (Figure 1D). Therefore, we hypothesized that CMC, as a dispersant, does not affect diet-induced obesity. To test this hypothesis, we evaluated the effects of CMC on obesity and eWAT inflammation in HFD-fed mice. Consistent with the results shown in Figure 1D, CMC administration did not enhance the relative abundance of either species, even in HFD-fed mice (Figure 5A,B). Thus, CMC did not impact HFD-induced body weight gain, adipose tissue weight, or appetite (Figure 5C–E). Furthermore, CMC did not alter expression levels of TNFa and CD11c in adipose tissues or colonic *Tjp1* expression; however, CMC decreased the expression level of F4/80 without affecting that of CD11c (Figure 5F,G). These results indicate that CMC, rather than PEG400, may be a preferable dispersant, at least for developing drugs targeting metabolic diseases. In fact, compared with CMC, PEG400 significantly increased gut *P. goldsteinii* even in the absence of isoliquiritigenin (ILG) (Appendix A), a polyphenol that exerts an anti-obesity effect by increasing beneficial bacteria [20]. Thus, PEG400 obscured the effect of ILG on the abundance of *P. goldsteinii* when ILG was dispersed in PEG400 for oral administration (Appendix A).

## 4. Discussion

It was reported that 50% PEG400 administration does not impact body weight in rodents fed a normal chow diet [32]. In contrast, we demonstrated that administering a high concentration of PEG400 could ameliorate the HFD-induced body weight gain (Figure 2), and this effect was, at least in part, attributed to PEG400-associated gut bacteria (Figure 4). Therefore, PEG-associated bacteria could ameliorate body weight under over-nutrition conditions.

Typically, the molecular weight of PEG can impact its absorption efficiency when orally administered [33]. Herein, mice administered 40% PEG4000 showed a greater reduction in α-diversity than those administered 40% PEG400 (Figure 1A), possibly reflecting the difference in PEG400 and PEG4000 contents reaching the large intestine lumen. Conversely, β-diversity values were comparable between the two groups (Figure 1B), indicating the presence of quantitative but not qualitative differences between PEG400- and PEG4000-associated bacteria. Therefore, we explored the effects of PEG400-associated bacteria on HFD-induced obesity in subsequent experiments (Figure 2, Figure 3 and Figure 4).

Regarding PEG400-associated bacteria, we found that *A. muciniphila* and *P. goldsteinii*, both anti-metabolic disease-associated species, were overrepresented in ND- and HFD-fed mice (Figure 1D and Figure 2C). Both species were shown to improve obesity and diabetes, partly by improving adipose tissue inflammation and gut barrier function [11,18]. Considering that FMT from PEG-administered mice ameliorated HFD-induced body weight and adipose tissue weight gain, as well as adipose tissue inflammation (Figure 4), changes in the gut microbiota could underlie the PEG-mediated effects against metabolic diseases.

Despite improved adipose tissue inflammation and overrepresentation of *A. muciniphila* and *P. goldsteinii*, PEG400 administration failed to improve insulin resistance by upregulating *Tjp1*, which encodes the tight junction protein ZO-1 and is implicated in gut barrier function [31]. Orally administered PEG was shown to increase the blood concentration of orally administered 4 kDa FITC-dextran [1], indicating that PEG400 could enhance gut permeability. Although the mechanism underlying PEG-induced leaky gut remains unknown, PEG may counteract the induction of tight junction genes/proteins by beneficial bacteria in colonic epithelial cells. Conversely, PEG was shown to enhance colonic barrier function and ameliorate chemically induced colitis [34]; the authors demonstrated a beneficial effect on gut integrity using 6% PEG4000, which was markedly less than that used in the current and previous studies, showing no or negative effects on gut barrier function. Therefore, the effect of PEG on the gut barrier function and gut bacterial composition may depend on its concentration.

One unresolved issue is the mechanism through which PEG increases *A. muciniphila* and *P. goldsteinii* abundance. A possible mechanism is the direct effect of PEG on bacterial growth, given that previous studies reported that PEG in culture medium promotes bacterial growth [35] (e.g., *Streptococcus* and *Enterococcus* spps.). Some bacterial species contain enzymes capable of degrading PEG, thus utilizing PEG as a carbon source [6]. Although the effect of PEG on the growth of bacteria, particularly anaerobic bacteria that abundantly inhabit the colonic lumen, remains unexplored, PEG may act as a carbon source or growth accelerator for *A. muciniphila* and *P. goldsteinii*.

Another possible mechanism is the indirect effect of PEG on bacterial composition by altering the gut microenvironment. The composition of the gut microbiota is altered by the quality and quantity of secretory IgA produced from plasma cells, as well as mucin secretion from goblet cells [36]. IgA binds to both pathogenic and non-pathogenic bacteria, contributing to the maintenance of gut homeostasis and regulating the activity and composition of gut commensal bacteria [37]. Cells associated with IgA secretion (e.g., B cells, plasma cells, and dendritic cells) are present in gut-associated lymphoid tissues, such as Peyer’s or cecal patches [38]. Reportedly, activation of Dectin-1, expressed in dendritic cells and receptors for sugar chains of polysaccharides such as b-glucan, can induce IgA secretion in the intestinal mucosa [39]. Although direct evidence is lacking, we speculate that PEG might modify the immune cell function associated with IgA secretion, resulting in altered gut microbiota diversity and composition.

Mucin is an energy source for *A. muciniphila*, which inhibits the invasion of pathogenic bacteria [40]. Previously, we demonstrated that ILG, a licorice-derived flavonoid polyphenol, could enhance the abundance of *A. muciniphila* in vivo [20]. The overrepresentation of *A. muciniphila* may be partly explained by the direct effect of ILG on mucin induction, which was confirmed in feces collected from ILG-fed mice and LS174T goblet-like cells treated with ILG; however, the detailed mechanism underlying mucin induction by polyphenols, including ILG, remains unknown [41]. Therefore, PEG400 administration might increase the abundance of *A. muciniphila* by impacting mucin secretion.

*P. goldsteinii* received considerable attention as a novel probiotic, and factors impacting its growth were identified [11]. To date, only polysaccharides isolated from *Hirsutella sinensis* and ILG were shown to increase gut *P. goldsteinii* abundance [11,20]; however, the underlying mechanism remains unknown. The mechanism underlying the PEG400-mediated changes in bacterial composition needs to be elucidated in future studies.

Clinically, PEG400 is currently used for pharmaceutical formulation; however, its amount in the drug is not abundant. Given that a high amount of PEG400 is not applied to clinical use and that its low dose did not markedly alter gut microbiota composition in mice, its influence on human gut microbiota in the context of clinical application may be minimal. In contrast, in animal experiments, an alternative dispersant lacking anti-obesity effects should be selected. Here, to explore the candidate dispersant, mice were challenged with long-term CMC administration. Oral administration of CMC did not ameliorate diet-induced obesity, which corresponded to a minimal change in gut microbiota diversity and composition when compared to the effects of PEG400 (Figure 1C and Figure 5). CMC is a water-soluble cellulose derivative with carboxymethyl groups bound to the hydroxyl groups [42]. Cellulose, the most abundant fiber in plants, consists of D-glucose residues linked by beta-1,4-linkage and is well known as a low-fermentable fiber for gut commensal bacteria [43], minimally altering the gut microbial community when administered (although a diet containing considerable (30%) cellulose was recently reported to substantially modify gut microbiota composition [44]). Considering that the backbone of CMC is low-fermentable cellulose, it is plausible that oral CMC administration did not markedly impact the gut bacterial community. Our results suggest that CMC is a preferable dispersant over PEG400, at least when exploring metabolic diseases.

In conclusion, we, for the first time, demonstrated the anti-obesity effect of PEG mediated via the modification of gut bacteria. We propose that the influence of gut microbiota should be considered when selecting dispersants for animal experiments.

## Figures and Tables

**Figure 1 microorganisms-11-01882-f001:**
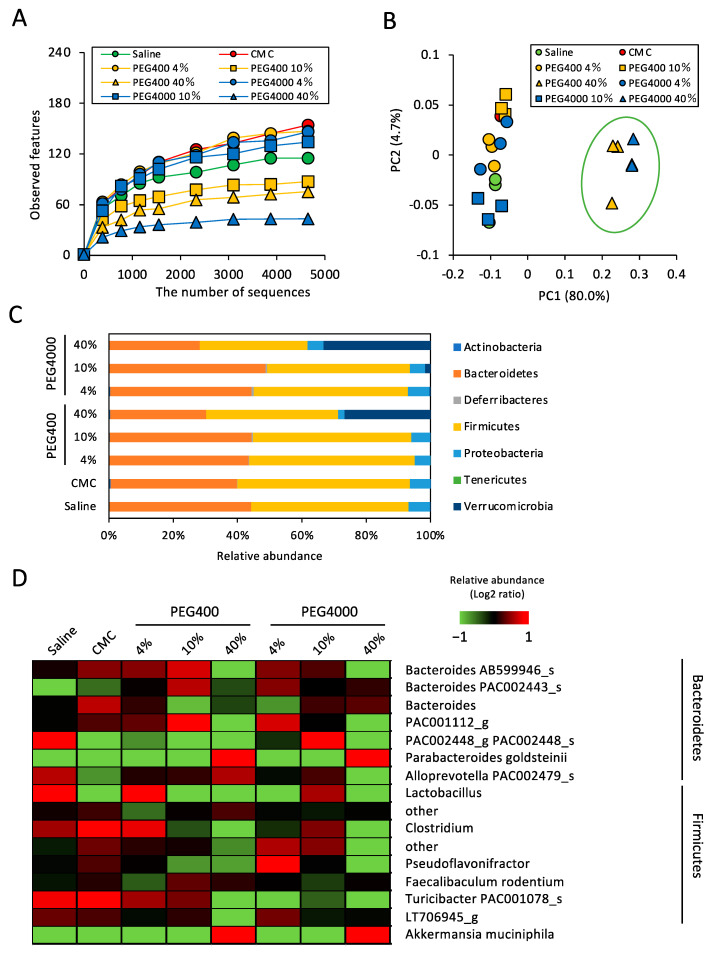
Effect of PEG on gut bacterial diversity and composition in mice fed a normal diet. (**A**) The effects of PEG or CMC on the α-diversity (observed features) of the gut bacteria analyzed by 16S rRNA sequencing (*n* = 3). (**B**) Principal component analysis of β-diversity (weighted UniFrac) values (*n* = 3). (**C**) Effect of PEG or CMC on phylum level gut bacterial composition (*n* = 3). (**D**) Relative abundance of the gut bacteria at the genus and species levels in mice administered saline, PEG, or CMC (*n* = 3). A mean abundance of more than 1% was extracted. PEG, polyethylene glycol; CMC, carboxymethyl cellulose.

**Figure 2 microorganisms-11-01882-f002:**
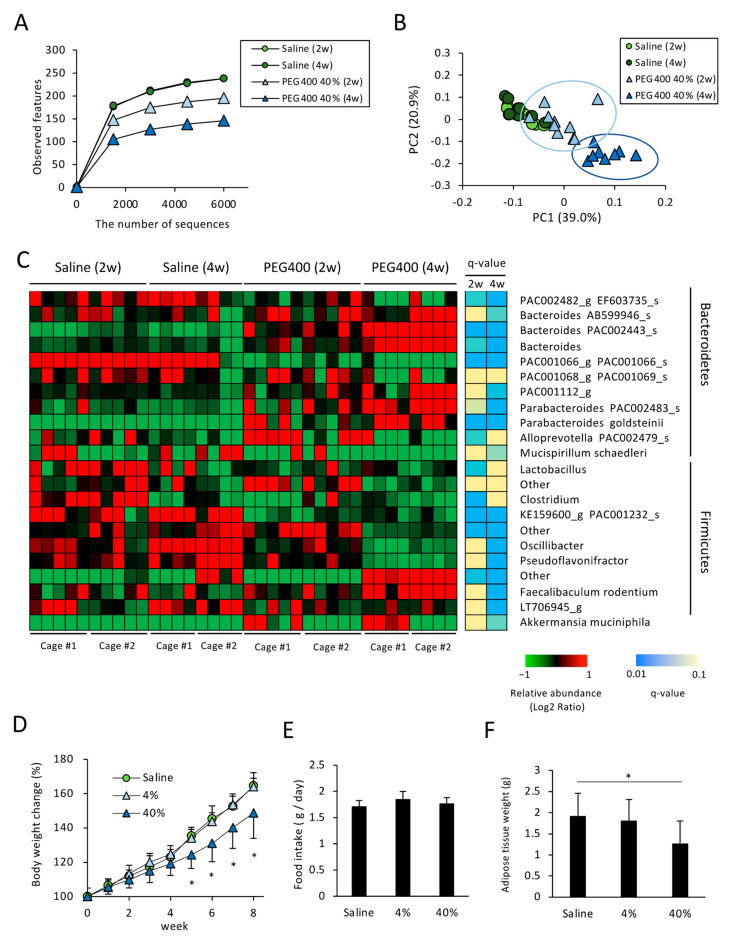
Effect of PEG400 on gut bacterial composition and diet-induced obesity in HFD-fed mice. (**A**) α-Diversity values (observed features) for gut bacteria in HFD-fed mice administered PEG400 for two or four weeks (*n* = 8–10). (**B**) Principal component analysis of β-diversity (weighted UniFrac) values (*n* = 8–10). (**C**) Relative abundance of the gut bacteria at the genus and species levels in mice administered with saline or PEG400 for two or four weeks (*n* = 8–10). A mean abundance of more than 1% was extracted. (**D**–**F**) Effect of PEG400 administration on body weight change, food intake, and adipose tissue weight (*n* = 10). Values and error bars indicate mean ± standard deviation (SD). * *p* < 0.05. HFD, high-fat diet; PEG400, polyethylene glycol 400.

**Figure 3 microorganisms-11-01882-f003:**
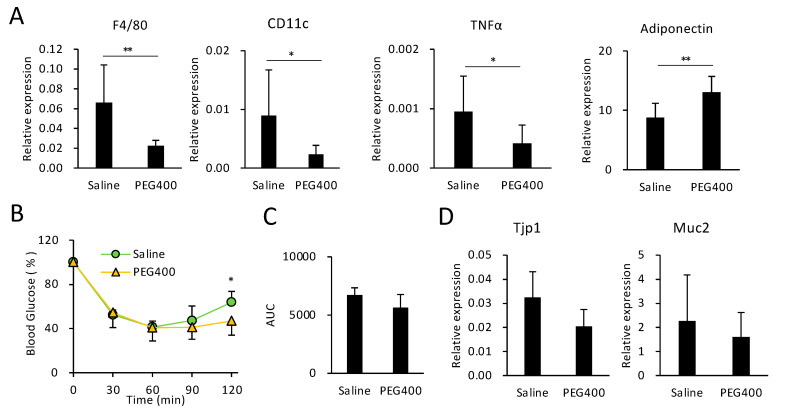
Effect of PEG400 on HFD-induced adipose tissue inflammation, insulin tolerance, and dysfunction of gut barrier integrity. (**A**) RT-qPCR analysis of inflammatory and adipokine-related genes in mouse epididymal white adipose tissue (eWAT) *(n* = 6–10). (**B**,**C**) Effect of PEG400 administration on insulin resistance (*n* = 6–10). (**D**) Effect of PEG400 on *Tjp1* and *Muc2* expression in mouse colonic tissue (*n* = 10). Values and error bars indicate mean ± standard deviation (SD). * *p* < 0.05; ** *p* < 0.01. HFD, high-fat diet; PEG400, polyethylene glycol 400; and RT-qPCR, reverse transcription-quantitative PCR.

**Figure 4 microorganisms-11-01882-f004:**
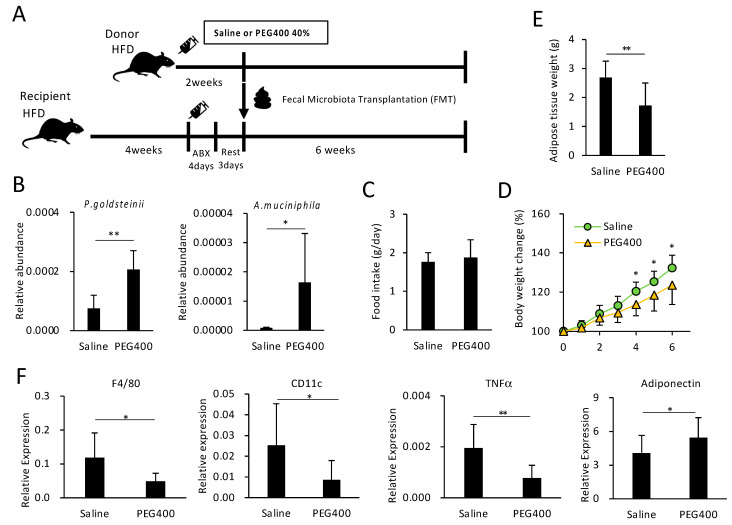
Effect of PEG400-modified gut bacterial transplantation on HFD-induced obesity and inflammation. (**A**) Schematic representation of study design for FMT following antibiotic (ABX) treatment. (**B**) Effect of FMT on the relative abundance of *Parabacteroides goldsteinii* and *Akkermansia muciniphila* in recipient feces. (**C**–**E**). Effect of FMT on food intake, recipient body, and adipose tissue weights (*n* = 9–10). (**F**) Effect of FMT on gene expression in recipient adipose tissue (*n* = 9–10). Values and error bars indicate mean ± standard deviation (SD). * *p* < 0.05; ** *p* < 0.01. FMT, fecal microbiota transplantation; HFD, high-fat diet; and PEG400, polyethylene glycol 400.

**Figure 5 microorganisms-11-01882-f005:**
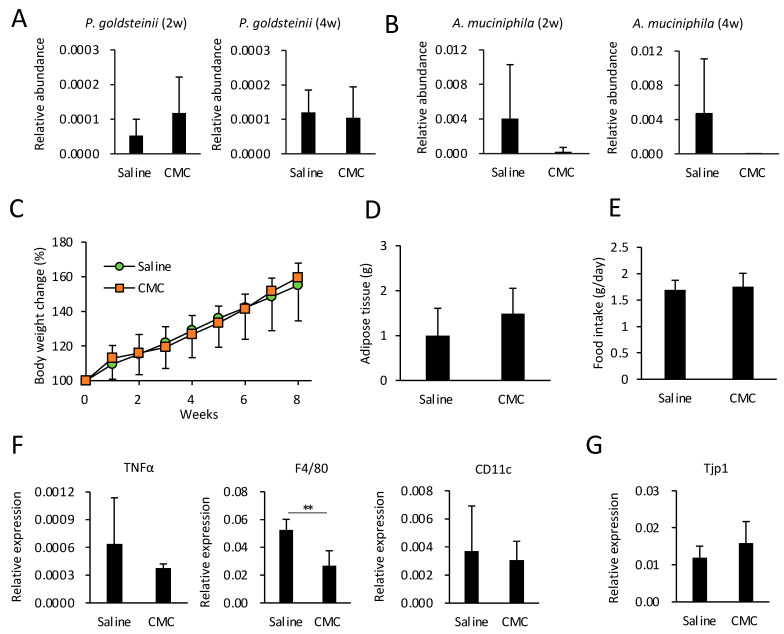
Effect of CMC administration on gut bacteria, diet-induced obesity, and adipose tissue inflammation in HFD-fed mice. (**A**) Effect of CMC on the relative abundance of gut *Parabacteroides goldsteinii* and *Akkermansia muciniphila* in HFD-fed mice. CMC was orally administered for two or four weeks, followed by the collection of fecal pellets (n = 6–7). (**B**–**E**) Effect of CMC on body weight, adipose tissue weight, and food intake (*n* = 6–7). (**F**) RT-qPCR analysis of inflammatory genes in eWAT of HFD-fed mice administered CMC (*n* = 6–7). (**G**) *Tjp1* expression levels in colonic tissue of HFD-fed mice administered CMC (*n* = 5–6). Values and error bars indicate mean ± standard deviation (SD). ** *p* < 0.01. CMC, carboxymethyl cellulose; eWAT, epididymal white adipose tissue; HFD, high-fat diet; and RT-qPCR, reverse transcription-quantitative PCR.

## Data Availability

Not applicable.

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
