# Peer review of "Effect of Oral Administration of Polyethylene Glycol 400 on Gut Microbiota Composition and Diet-Induced Obesity in Mice"

_microorganisms, 2023, doi:10.3390/microorganisms11081882_

Round 1

Reviewer 1 Report

Polyethylene glycol 400 (PEG 400) is a widely used compound with excellent biocompatibility and low toxicity. This study aimed to investigate the effect of oral administration of PEG 400 on gut microbiota composition and its potential impact on diet-induced obesity in mice. The meaning of this finding will have to be more deeply considered in future microbiota composition research. 

Comment1: The manuscript is generally lacking in methodology details and clarity – especially relevant for the design of target bacteria primers used for qPCR. Please explain how primer specificity was evaluated.

Comment2: How could the microbiota present be affected by different antibiotic treatments, and how might this subsequently influence the reaction to the antibiotic?

Comment3: " 21. Yamanouchi, Y.; ...2022”. Check format of this reference - article page number is missing. Please make sure the reference list set in the same format. The Ethics Committee approval number should be added in the "2.1. Animals and Treatments" section to ensure compliance with ethical standards in the study.

Comment4: Schematic illustration for the effect of oral administration of polyethylene glycol 400 on gut microbiota composition and diet-induced obesity in mice would be helpful to non-expert readers.

Comment5: Revise abstract so it is less general and does not overstate manuscripts main conclusion. Abstract should be revised and describe specific experiments done and results of these experiments. In addition to that, to increase the depth of the research article, conducting additional Western Blot (WB) experiments for the specific target gene can be beneficial.

Author Response

Response to Reviewers’ comments

We are grateful to the reviewers for the critical comments and helpful suggestionsthat have improved the overall quality of our paper. We have taken almost all these comments and suggestions into account while revising our manuscript. Below is a point-by-point response to the issues raised. For each specific point, the reviewer’s comments are in bold-italicsand our responses are in plain type-font.

Reviewer #1

Comment1: The manuscript is generally lacking in methodology details and clarity – especially relevant for the design of target bacteria primers used for qPCR. Please explain how primer specificity was evaluated.

We apologize for the lack in the details of the methodology. We have revised the Material and Methods section (section 2; highlighted in green).

Regarding the primers for bacterial species, we used primer sets from our previous study [reference #20]. Primer specificity was previously evaluated by both computational homology search using NCBI BLAST and agarose gel electrophoresis post PCR amplification.

Comment2: How could the microbiota present be affected by different antibiotic treatments, and how might this subsequently influence the reaction to the antibiotic?

Comparison of changes in gut microbiota composition following treatment with four individual antibiotics has recently been reported (DOI: 10.1128/spectrum.01904-21). As shown in this report, different antibiotic treatments result in different gut microbiota composition since the antimicrobial spectrum is dependent on the type of antibiotic. However, in our experiment, a cocktail containing the four antibiotics instead of individual antibiotics was used to broadly deplete gut microbiota (referenced from DOI: doi:10.3791/58481) for FMT. In terms of the presence of gut microbiota, the total amount of DNA extracted from feces decreased following administration of the antibiotics, indicating that antibiotics treatment decreases the abundance of gut bacteria.

Regarding the subsequent influence on the reaction to the antibiotics, Huang et al. showed that long-term antibiotic treatment promoted the development of antibiotic-resistant bacterial strains in the gut (DOI: 10.1128/spectrum.01904-21). Therefore, repeated antibiotic treatment might lead to the resistance to same antibiotic for certain types of bacterial strains. However, the late effects were not clarified in our experiment, which performed antibiotics cocktail treatment for only 4 d.

Comment3: " 21. Yamanouchi, Y.; ...2022”. Check format of this reference - article page number is missing. Please make sure the reference list set in the same format. The Ethics Committee approval number should be added in the "2.1. Animals and Treatments" section to ensure compliance with ethical standards in the study.

Thank you for your valuable suggestion. The page number was missing for references #21, #31, #32, and #43. We have added the page number in the revised manuscript. In addition, we have included the Ethics Committee approval number in Section 2.1 of the revised manuscript (lines 86–88, highlighted in yellow).

Comment4: Schematic illustration for the effect of oral administration of polyethylene glycol 400 on gut microbiota composition and diet-induced obesity in mice would be helpful to non-expert readers.

Following the reviewer’s helpful comment, we have added a simple graphical abstract to summarize the results of this study.

Comment5: Revise abstract so it is less general and does not overstate manuscripts main conclusion. Abstract should be revised and describe specific experiments done and results of these experiments. In addition to that, to increase the depth of the research article, conducting additional Western Blot (WB) experiments for the specific target gene can be beneficial.

Thank you for your helpful suggestion. We have corrected abstract such that we would not overstate the main conclusion of our study. Specifically, we changed the following sentence:

[Previous version]

In conclusion, we demonstrated that oral administration of PEG400 could ameliorate diet-induced obesity by altering the gut microbiota composition, thereby indicating that the type of dispersant should be carefully considered in animal experiments.

[Current version] (lines 26­–28)

In conclusion, we demonstrated that oral administration of a high concentration of PEG400 (40%) alters the gut microbiota composition and ameliorates HFD-induced obesity.

Moreover, as the reviewer indicated, addressing the protein expression level will further support the conclusion of this study. However, we have not collected samples for WB from HFD-fed mice administered PEG400. It will take more than 8 weeks for us to prepare PEG400-treated mice for WB analysis. We will collect samples and perform WB in a future study comparing the effect of several dispersants including PEG400 (currently in the planning stage).

Reviewer 2 Report

The paper "Effect of oral administration of polyethylene glycol 400 on gut microbiota composition and diet-induced obesity in mice" reports about an experimental work on the effect on diet-induced obesity of Polyethylene glycol commonly used as a dispersant for oral administration of hydrophobic agents. Effects of PEG are also investigated. The paper is well written, experiments seem properly performed and results are convincing. However, some parts of the text need to be improved, as detailed below, to make the paper suitable for publication.

Figure 1 shows the comparative results on the effects of different connentration of PEG 400 and 4000 and of CMC on gut bacteria, and at lines 178-184 are practivally given the reasons for the choice of the bacterial phylum and PEG doses for the following experiments. A comment should be  given also on CMC and the absence of remarkable changes as compared to saline, directing thus the reader to section 3.5

Lines 305-307. The comment on P. goldsteinii, reported as an anti-metabolic disease-associated species, is to be revised according to data shown in  Supplementary Fig. 1, where it seems that as compared with CMC, PEG400 is able to improve P. goldsteinii also in the absence of ILG.

Given the title of the Special Issue (Gut Microbiota: Health, Clinical & Beyonds) and the implication of the dispersants on drug administration in clinical, a sentence on the translation of this study to the clinics and possible limitations would be welcome.

Minor remarks

Figure 1A, and related texts, please give a clear explanation of “observed features”

Good

Author Response

Response to Reviewers’ comments

We are grateful to the reviewers for the critical comments and helpful suggestionsthat have improved the overall quality of our paper. We have taken almost all these comments and suggestions into account while revising our manuscript. Below is a point-by-point response to the issues raised. For each specific point, the reviewer’s comments are in bold-italicsand our responses are in plain type-font.

Reviewer #2

Figure 1 shows the comparative results on the effects of different connentration of PEG 400 and 4000 and of CMC on gut bacteria, and at lines 178-184 are practivally given the reasons for the choice of the bacterial phylum and PEG doses for the following experiments. A comment should be given also on CMC and the absence of remarkable changes as compared to saline, directing thus the reader to section 3.5 

Thank you for your helpful suggestion. We have added a statement indicating that bacterial composition at the phylum level was comparable between mice administered CMC and those administered saline (lines 190–191 in the revised manuscript, highlighted in yellow)

Lines 305-307. The comment on P. goldsteinii, reported as an anti-metabolic disease-associated species, is to be revised according to data shown in  Supplementary Fig. 1, where it seems that as compared with CMC, PEG400 is able to improve P. goldsteinii also in the absence of ILG.

As the reviewer pointed out, PEG400 increased the level of P. goldsteiniiin the absence of ILG. This may be an adverse side effect of PEG400 in animal experiments wherein the effect of drugs on metabolic disease is being evaluated. To emphasize this point, we have revised this sentence as follows (lines 315–319 in the revised manuscript, highlighted in yellow):

[Previous version]

In fact, compared with CMC, PEG400 obscured the significant increase of P. goldsteiniiwhen isoliquiritigenin (ILG), a polyphenol that has an anti-obesity effect by increasing beneficila bacteria [26], was dispersed for oral administration (Supplementary Fig. 1)

[Current version]

In fact, compared with CMC, PEG400 significantly increased gut P. goldsteiniieven in the absence of isoliquiritigenin (ILG) (Supplementary Fig. 1), a polyphenol that exerts an antiobesity effect by increasing beneficial bacteria [26]. As a result, PEG400 obscured the effect of ILG on the abundance of P. goldsteinii when ILG was dispersed in PEG400 for oral administration (Supplementary Fig. 1)

Given the title of the Special Issue (Gut Microbiota: Health, Clinical & Beyonds) and the implication of the dispersants on drug administration in clinical, a sentence on the translation of this study to the clinics and possible limitations would be welcome.

Thank you for your suggestion. Given that a low dose of PEG400 did not affect mouse gut microbiota composition and that a high amount of PEG400 is not orally administered for clinical patients, the influence of PEG400 on human intestinal bacteria may be minimal. We have added a statement for this in the revised manuscript (lines 402–407, highlighted in yellow).

Minor remarks 

Figure 1A, and related texts, please give a clear explanation of “observed features”

“Observed features” is one of the calculated parameters explaining richness (the total number of species in a community). Chao1 was previously used to show richness in QIIME1, whereas “Observed features” is currently adopted in QIIME2. We have added a description of “Observed features” in the revised manuscript (lines 116–117, highlighted in yellow)

Round 2

Reviewer 1 Report

The authors answered all my questions and the manuscript has been much improved.